# Role of Iron and Iron Overload in the Pathogenesis of Invasive Fungal Infections in Patients with Hematological Malignancies

**DOI:** 10.3390/jcm11154457

**Published:** 2022-07-30

**Authors:** Toni Valković, Marija Stanić Damić

**Affiliations:** 1Department of Hematology, University Hospital Rijeka, Krešimirova 42, 51000 Rijeka, Croatia; 2Faculty of Medicine Rijeka, University of Rijeka, Braće Branchetta 20, 51000 Rijeka, Croatia; 3Faculty of Health Studies, University of Rijeka, Viktora Cara Emina 5, 51000 Rijeka, Croatia

**Keywords:** iron, iron overload, fungal infection, hematological malignancies, iron chelation therapy

## Abstract

Iron is an essential trace metal necessary for the reproduction and survival of fungal pathogens. The latter have developed various mechanisms to acquire iron from their mammalian hosts, with whom they participate in a continuous struggle for dominance over iron. Invasive fungal infections are an important problem in the treatment of patients with hematological malignancies, and they are associated with significant morbidity and mortality. The diagnosis of invasive clinical infections in these patients is complex, and the treatment, which must occur as early as possible, is difficult. There are several studies that have shown a possible link between iron overload and an increased susceptibility to infections. This link is also relevant for patients with hematological malignancies and for those treated with allogeneic hematopoietic stem cell transplantation. The role of iron and its metabolism in the virulence and pathogenesis of various invasive fungal infections is intriguing, and so far, there is some evidence linking invasive fungal infections to iron or iron overload. Clarifying the possible association of iron and iron overload with susceptibility to invasive fungal infections could be important for a better prevention and treatment of these infections in patients with hematological malignancies.

## 1. Invasive Fungal Infections in Hematology

Despite some effective prophylaxis modalities, invasive fungal infections (IFI) caused by Aspergillus and Candida species, and more rarely by Zygomicetes, Fusarium or Trichosporon species, are still a common cause of morbidity and mortality in immunocompromised patients with hematological malignancies, including those who are allogeneic hematopoietic stem cell transplant recipients [1,2]. The main cause of a predisposition for IFI is the impairment of immunity that results from the pathogenesis of the malignant hematological disease itself, but also from various therapies that weaken immunity at different levels. Crucial to IFI susceptibility is an impaired innate immunity (a reduced number and function of neutrophils and macrophages), and also an impaired T cell immunity. This is why the incidence of IFI is highest in patients with acute leukemia treated with intensive chemotherapy or those who have undergone allogeneic bone marrow transplantation. The main risk factor for the development of IFI in hematological patients is a severe and prolonged neutropenia after intensive chemotherapy [3]. In hematopoietic cell transplant recipients, there are three main factors that increase the risk of IFI: mucosal damage and neutropenia as an early consequence of transplantation, severe damage and gradual recovery of T-cell immunity, and prolonged corticosteroid treatment in patients who develop acute graft-versus-host disease [4,5,6]. There are several papers that investigated the role of IFI in the hematological patients unfit for chemoimmunotherapy but treated with other emerging therapies. Aldoss et al. concluded that the overall risk of IFI during venetoclax and hypomethylating agent therapy is relatively low (12.6% patients developed probable or proven IFI in this investigation). The risk of IFI was higher in nonresponders, relapsed and refractory patients [7]. There are real-world data suggesting a slightly higher risk of IFI in patients with chronic lymphocytic leukemia treated with ibrutinib (BTK inhibitor), especially in the first six months of treatment [8,9,10]. 

The same could be true for PI3K inhibitors such as idelalisib [8,10,11]. All this suggests the possible importance of antifungal prophylaxis in certain groups of patients. In principle, posaconazole remains the drug of choice when the incidence of invasive mold diseases exceeds 8%, and it is strongly recommended for patients undergoing remission-induction chemotherapy for acute myeloid leukemia and myelodysplastic syndrom, as well as for preventing IFI in recipients of an allogenic hematopoietic stem cell transplantation, especially post-engraftment in the presence of graft-versus-host disease and other risk factors for IFI [12].

The diagnosis of IFI must be fast and effective, and it is crucial to start a specific antifungal therapy as early as possible, as this is the most important prerequisite for successful treatment and prevention of mortality [13]. The diagnosis of IFI itself is established by means of a combination of different diagnostic methods: microbiological cultures, microscopy, various antibody/antigen tests, molecular and imaging diagnostics. The basis of prophylaxis and forehand treatment of IFI are different antifungal drugs (triazoles, echinocandins and polyenes), as well as surgical treatment in some cases.

## 2. Iron and Iron Overload: Their Role in Infections

Iron, a micronutrient essential for life, participates in many vital biological processes. Ferritin is a protein that serves to store iron in the body in a non-toxic form. It is mainly located intracellularly, although a small proportion of this protein is found in the serum and correlates with the body’s total iron stores [14]. When the concentration of ferritin in the body (serum) is higher than normal, we talk about iron overload. The latter can result from various pathological conditions, such as primary hemochromatosis or frequent blood transfusions, which are common in some hematological diseases. Elevated serum ferritin levels also occur with infections or liver damage [15]. Like humans, microorganisms also need iron for their growth and survival. Thus, fungal pathogens have developed various mechanisms to obtain iron from hosts for their own needs, which will be discussed in more detail in the following section. Hosts, including humans, on the other hand, have developed mechanisms to make iron as inaccessible to microorganisms as possible, especially during infection and concomitant inflammation (increased production of hepcidin and natural iron chelators being some of them), which probably reduces the virulence and pathogenicity of bacterial and fungal pathogens [16]. Therefore, there is a constant competitive relationship between hosts and microorganisms, including fungal pathogens, with respect to iron. 

While it is well known that iron overload damages organs such as the liver, the heart, and the endocrine glands, the impact of this disorder on the immune system is less well understood (Figure 1). Although the diagnosis of iron overload can be confirmed by various invasive and non-invasive (imaging) methods, the simplest, most widely used, but not the most accurate method is to determine serum ferritin values; a serum ferritin value of 1000 µg/L or more is mainly used in hematology as a cut off value, which indicates treatment with iron chelators [17,18]. In their meta-analysis, Oliva and colleagues suggested that a higher concentration of serum ferritin is a prognostic indicator of shorter survival in patients with myelodysplastic syndrome, as shown by some earlier research [19,20]. Although some data suggest that non-transferrin-bound iron and labile plasma iron could have a proleukemic effect achieved through reactive oxygen species (ROS) [21,22], the two papers included in this meta-analysis did not establish a possible relationship between serum ferritin level and progression to AML [23,24]. There is evidence that iron overload stimulates the growth and survival of some microorganisms. For example, Vibrio vulnificus and Yersinia enterocolitica, the so-called “siderophylic bacteria”, have been shown to cause dangerous infections, particularly in patients with iron overload [25,26,27]. Furthermore, macrophage iron overload due to chronic hemolysis in malaria has been proven to increase the risk of Salmonella infections [28]. Finally, several authors have shown that iron supplementation leads to increased morbidity and mortality from different endemic infections [29,30]. The results of a number of studies found that elevated serum ferritin or hepcidin-25 was associated with more frequent infections in hemodialysis patients and in those who underwent kidney or liver transplantation [31,32,33]. In hematology, several studies linked iron overload with a higher incidence of infections. Pre-transplantation ferritin values have been shown to be positively correlated with the incidence of bloodstream infections within 100 days of allogeneic bone marrow transplantation [34], as well as early bacterial infections after transplantation [35]. In studies of multiple myeloma, Miceli and colleagues found that iron overload is a significant risk factor for infections after autologous transplantation in patients with this type of cancer [36]. Our group also found that elevated serum ferritin is an important risk factor in patients with multiple myeloma who did not undergo transplantation [37].

In affected organs, excess iron can chemically interact with hydrogen peroxide, creating reactive oxygen species that can cause tissue damage, inflammation, and fibrosis. As such, iron overload can lead to cardiomyopathy, arrhythmias and heart failure, liver fibrosis and cirrhosis, diabetes mellitus, hypothyroidism, hypogonadism, and impotence. The impact of iron overload on the immune system and infections is less well understood, although there are indications that iron overload stimulates the growth and survival of some microorganisms and that excessive amounts of iron in bone marrow stores act as an independent prognostic factor for invasive aspergillosis in allogenic transplant patients.

## 3. Mechanisms of Iron Acquisition by Fungal Pathogens

Like bacteria, fungal pathogens need iron for their survival, as this micronutrient participates in important biological processes such as DNA replication, transcription, metabolism, and energy generation, which is especially relevant during infection, when both the host and the fungal pathogen are struggling to survive [38]. As we shall see below, there are several different mechanisms that fungal pathogens have developed in order to obtain iron from their mammalian hosts.

### 3.1. Reduction of Ferric to Ferrous Iron with Subsequent Transport

Yeast *S. cerevisiae* was used to investigate the basic mechanisms involved in the acquisition of iron by fungal pathogens. The reduction of ferric to ferrous iron with subsequent transport takes place in two stages: in order to enter the cell, the insoluble ferric iron must first be reduced to a relatively soluble ferrous iron, which is mediated by ferric reductases encoded by FRE genes (FRE1 and FRE2). The second stage consists of the re-oxidation of ferrous iron to the ferric form, which is accomplished by multicopper ferroxidase (Fet3) coupled with transport into the cell by a permease (Ftr1) [39]. This process is necessary because the ferrous form is toxic to the cell as it leads to the formation of reactive oxygen species. 

*C. albicans*, *A. fumigatus* and *C. neoformans* use the same cell-surface-mediated ferric reductases, ferroxidases and iron permeases as described in *S. cerevisiae* [40,41,42,43,44]. Genes for other proteins that play a role in ferric reductases have been found in the genome of *C. albicans* and *A. fumigatus*, although probably not all of them are active [42,43,45,46]. 

### 3.2. Siderophore Production and Transport

Most fungal pathogens can produce and secrete siderophores, which are tiny organic formations whose most notable function is to be high affinity ferric chelators that acquire and transport iron from the microenvironment into the interior of the cell [47,48,49,50]. The *Aspergillus* species have the capacity to synthesize several types of siderophores, including ferricrocin, hydroxyferricrocin, fusarinine C, coprogen B and triacetylfusarinine C [51,52,53]. There are plenty of studies which looked at the genetics and production of siderophores, as well as their possible role in the virulence of *A. fumigatus* [54,55,56]. Despite the fact that a reductive transport system can perform a siderophore-related input, this system is most efficient in an abundance of siderophores. At lower concentrations of siderophores, entry into the cell occurs primarily through specific transporters of the ARN/SIT subfamily of the major facilitator superfamily, which are secondary transporters with 14 predicted transmembrane domains, and they likely function as proton symporters energized by the membrane potential [57,58,59]. 

*S. cerevisiae*, *C. neoformans* and *C. albicans* cannot produce their own siderophores. Instead, they use exogenous siderophores (xenosiderophores) synthesized by other microorganisms. The research on *S. cerevisiae* provided basic findings and a model for the uptake mechanisms for xenosiderophores via different transporters [57]. These transporters, such as Arn1, Arn2/Taf1, Arn3/Sit1, and Arn4/Enb1, are specific for different bacterial and fungal xenosiderophores, such as enterobactin, ferrichrome, ferrichrome A, triacetylfusarine C, and ferrioxmaine B [60]. As an example, *C. albicans* uses the Sit1/Arn1 transporter to uptake xenosiderophores such as ferricrocin, ferrichrysin, ferrirubin, coprogen and triacetyl-fusarine C [61].

### 3.3. Iron Acquisition from Host’s Iron-Containing Proteins Such as Hemoglobin and Other Proteins

The great majority of iron in mammalian hosts is contained in hemoglobin. A small amount of iron can also be found in other heme-containing proteins such as transferrin, haptoglobin, lactoferrin, haemopexin, lipocalin-1 and lipocalin-2 (Lcn1/Lcn2) [62]. In the human organism, intracellular iron is sequestrated from fungal pathogens thanks to the proteins transferrin and ferritin [62,63]. Fungal pathogens can also obtain the iron necessary for their survival during a mammalian host infection from heme, i.e., hemoglobin or other iron-containing proteins. This process requires access to host hemoglobin sources, so that fungal agents can produce hemolysins that lead to erythrocyte breakdown and hemoglobin release, or they can secrete proteases that degrade iron-containing proteins [38]. In general, microorganisms, including molds and fungi, possess two mechanisms for the acquisition of iron from heme and heme-containing proteins: direct heme uptake or uptake by hemophores, which represent heme-binding proteins. For example, *C. albicans* can obtain iron from heme/hemoglobin using hemoglobin as a key iron source [64,65,66]. It was previously discovered that *C. albicans*, which possesses the ability to perform hemolysis, binds erythrocytes through complement receptor-like molecules [67]. The further uptake of hem/hemoglobin is mediated by specific receptors on the surface of *C. albicans*, including the conserved family of proteins-Rbt5, Rbt51/Pga10, Pga7 and Csa2, which contain the cysteine-rich common in fungal extracellular membrane (CFEM) domain [64,65,68,69,70,71]. Furthermore, C. *neoformans* can produce a 43 KDa serine proteinase that degrades hemoglobin [72]. It requires the ESCRT protein Vps23, as well as the mannoprotein Cig1, for iron acquisition from heme [73]. It seems that the mannoprotein Cig1 performs the function of a hemophore in *C. neoformans* and thus contributes to iron acquisition [74]. There is some evidence for fungal acquisition of iron from other iron-containing proteins. For example, C. *albicans* can utilize adhesin Als3 as a ferritin receptor with the ultimate goal of procuring iron for its growth and proliferation [75]. The different mechanisms of iron acquisition by the most common fungal pathogens are listed in Table 1.

## 4. Iron Overload, Iron Chelation Therapy and Invasive Fungal Infections in Hematology

In the literature to date, there are several studies on a small number of patients that have confirmed a positive association between altered iron metabolism, iron overload and fungal infections in hematological patients treated with allogeneic hematopoietic stem cell transplantation [76,77,78]. Kontoyiannis and colleagues found in a retrospective study of a large number of patients that excessive amounts of iron in bone marrow stores are an independent prognostic factor for invasive aspergillosis in allogenic transplant patients [79]. Alessandrino and colleagues found that pre-transplantation transfusion history and serum ferritin level (as a measure for iron overload) have a significant prognostic value in patients with myelodysplastic syndrome undergoing myeloablative allogeneic stem cell transplantation, inducing a significant increase in non-relapse mortality. Although fungal infections are a common cause of non-relapse mortality, the authors did not analyze their incidence in their cohort of patients [80]. There are several experimental and clinical papers that point out the positive effect of iron chelators on the treatment of fungal pathogens (Figure 2). For example, in the studies by Lee and colleagues and Chayakulkeeree and collegues, several iron chelators increased the efficacy of amphotericin B, showing synergism with this antifungal drug in the *Cryptococcus* strains [81,82]. A study by Ibrahim et al. showed that the iron chelator deferasirox protects the kidneys of mice from mucormycosis through iron starvation [83]. The same group demonstrated that the iron chelator deferasirox improves the efficacy of liposomal amphotericin B in murine invasive pulmonary aspergillosis [84]. 

Ye and colleagues showed that supplementation of calcium with an iron chelator increases antifungal drug efficacy against azole-resistant *A. fumigatus* isolates. They also demonstrated that calcium supplementation (calcium induces the downregulation of iron uptake-related genes and siderophore-mediated iron acquisition) combined with iron deficiency causes a dramatic growth inhibition of the human fungal pathogens *Aspergillus fumigatus*, *Candida albicans*, and *Cryptococcus neoformans* [85]. Furthermore, it seems that the pre-transplantation use of iron chelators could be an important step in reducing infectious complications related to allogeneic hematopoietic stem cell transplantation, including invasive fungal infections. For instance, Lee and colleagues showed in their research on pediatric patients that iron chelator therapy could be beneficial for improving the outcome of HSCT [86]. To confirm these preliminary findings, large prospective clinical studies should provide a definitive answer on whether iron chelator therapy could indeed reduce the incidence and severity of infection in other hematological diseases, not only in the context of HSCT.

## 5. Future Perspectives and Possible Therapeutic Application

In addition to therapy with iron chelators, which is an established method of treatment of hematological patients with iron overload, nowadays, there are numerous therapeutic attempts to treat bacterial and fungal infections by blocking the mechanisms by which microorganisms successfully acquire iron from the environment, i.e., their host. Amongst these models are the inhibition of siderophores’ metabolism (biosynthesis, secretion, import), the inhibition of uptake systems and biosynthetic enzymes, vaccines against surface uptake components and the utilization of siderophore conjugates to specifically deliver antibiotics/antimycotics to the pathogen [87,88,89,90,91,92]. These treatments are designed as additional and synergistic to standard antibiotics and antimycotics, especially in the treatment of highly resistant strains and in severe clinical cases.

In conclusion, it can be said that invasive fungal infections are still an important cause of mortality and morbidity in patients with hematological malignancies, especially those treated with intensive chemotherapy (acute leukemia and other diseases) and others who have undergone HSCT. Although environmental and other risk factors for this type of infection are well-documented, patients without risk factors also suffer from invasive fungal infections or, vice versa, those with known risk factors remain spared from the disease. This suggests that the pathogenesis of invasive fungal infections is still not sufficiently understood in hematological patients, and that knowledge of the mechanisms of iron acquisition and utilization by fungal pathogens, as well as new knowledge on the impact of iron overload on IFI, could be of great importance. 

The fight against microorganisms that cause infectious complications in patients with malignant hematological diseases is always difficult and uncertain, as the COVID-19 pandemic has painfully warned us. Although we currently have relatively effective drugs for the treatment of fungal infections, resistance to existing drugs and the emergence of new strains always pose threats. Thus, knowledge of other pathogenetic mechanisms that lead to increased virulence of fungal pathogens and invasive fungal infections is of great practical relevance. The use of iron chelators seems to be one of the already widely available therapeutic methods to reduce the availability of iron to pathogens, and it should not be underestimated and poorly used in the treatment of hematological patients and those undergoing HSCT who suffer from iron overload. However, various other therapeutic approaches based on blocking the mechanisms by which microorganisms successfully acquire iron from the environment must be further investigated in large clinical trials in order to become an additional treatment for IFI.

## Figures and Tables

**Figure 1 jcm-11-04457-f001:**
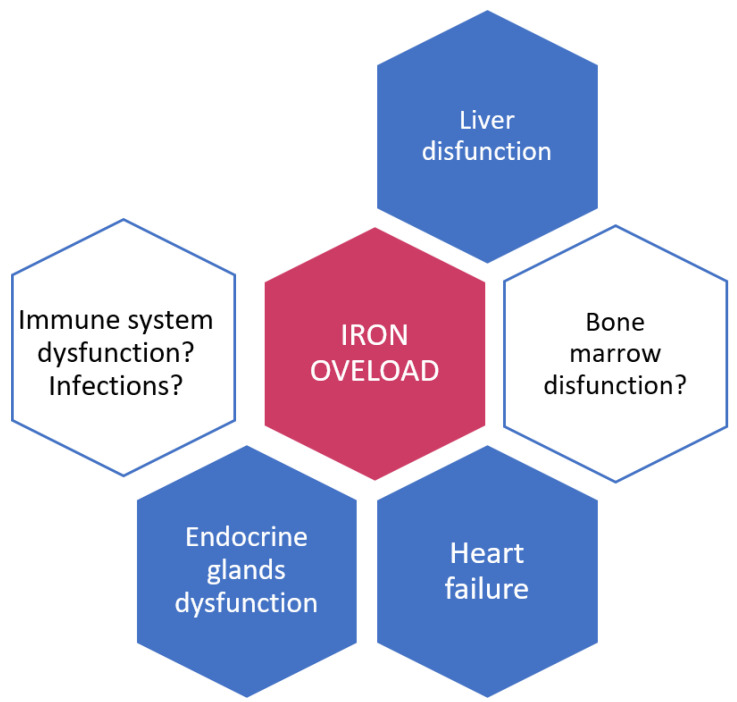
Effects of iron overload on target organs.

**Figure 2 jcm-11-04457-f002:**
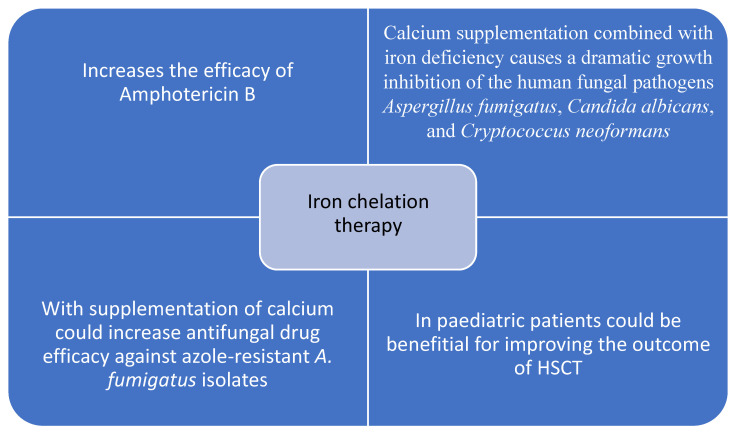
Possible effects of iron chelation on fungal pathogens. The use of iron chelators such as deferasirox increases the efficacy of the antifungal drug amphotericin B in invasive pulmonary aspergillosis [69,70,72] and could reduce infectious complications in pediatric patients undergoing HSCT [74]. Supplementation with calcium along with the use of iron chelators causes dramatic growth inhibition of the human fungal pathogens and increases efficacy of antifungal drugs [73].

**Table 1 jcm-11-04457-t001:** Mechanisms of iron acquisition by fungal pathogens.

*Cryptococcus neoformans*	❖ grows on hemoglobin and heme as sole iron sources [38] ❖ can produce 43 KDa serine proteinase that degrades hemoglobin [72] ❖ uses xenosiderophores synthesized by other microorganisms [57] ❖ reduces ferric to ferrous iron with subsequent transport [39]
*Saccharomyces cerevisiae*	❖ reduces ferric to ferrous iron with subsequent transport [39] ❖ uses xenosiderophores synthesized by other microorganisms via different transporters [57]
*Candida albicans*	❖ can obtain iron from heme/hemoglobin [64,65,66] ❖ can utilize adhesin Als3 as a ferritin receptor for procuring iron [75] ❖ uses xenosiderophores synthesized by other microorganisms, such as ferricrocin, ferrichrysin, ferrirubin, coprogen and triacetyl-fusarine C [61] ❖ possesses hemolytic activity and binds erythrocytes through complement receptor-like molecules [38,67] ❖ reduces ferric to ferrous iron with subsequent transport [39]
*Aspergillus fumigatus*	❖ synthesizes several types of siderophores such as fusarinine C (FsC)/triacetylfusarinine C (TAFC) and ferricrocin to obtain iron from transferrin [38,51,52,53] ❖ both intracellular and extracellular siderophores contribute to the virulence of A. fumigatus [38] ❖ the reductive iron uptake system does not play a role in virulence [38] ❖ reduces ferric to ferrous iron with subsequent transport [39]

## Data Availability

Not applicable.

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
