# Peer review of "Role of Iron and Iron Overload in the Pathogenesis of Invasive Fungal Infections in Patients with Hematological Malignancies"

_jcm, 2022, doi:10.3390/jcm11154457_

Round 1
Reviewer 1 Report
The manuscript submitted for review is well prepared and constitutes a substantively justified narrative literature review. Professor Valković is a well-known physician; I know his works in the field of MM.
I only have two minor comments:
1. tables and figures from supplementary materials should be included in the text of the work,
2. the text could be extended by a short section (literally a few sentences) about the role of iron overload in the pathophysiology and progression of hematological diseases (leukemia, MDS):
https://doi.org/10.3390/cancers13123029
https://doi.org/10.3390/jcm11030895
Author Response
Dear editor, thank you for your review.
we accepted your suggestions. We included tables and figures from supplementary materials in the text of the work and we extended the text by a short section about the role of iron overload in the pathophysiology and progression of hematological diseases (bolded in manuscript):
"In their meta-analysis, Oliva and colleagues suggest that a higher concentration of serum ferritin is a prognostic indicator of shorter survival in patients with myelodysplastic syndrome, as shown by some earlier research (1, 2). Although some data suggest that non-transferrin-bound iron and labile plasma iron could have a proleukemic effect achieved through ROS (reactive oxygen species) (3,4), the two papers included in this meta-analysis did not establish a possible relationship between serum ferritin level and progression to AML (5, 6)
References:
- Oliva EN, Huey K, Deshpande S et al. A Systematic Literature Review of the Relationship between Serum Ferritin and Outcomes in Myelodysplastic Syndromes. J Clin Med. 2022 Feb 8;11(3):895. doi: 10.3390/jcm11030895. PMID: 35160344; PMCID: PMC8836890.
- Malcovati L, Della Porta MG, Cazzola M. Predicting survival and leukemic evolution in patients with myelodysplastic syndrome. Haematologica. 2006 Dec;91(12):1588-90. PMID: 17145593.
- Weber S, Parmon A, Kurrle N, Schnütgen F, Serve H. The Clinical Significance of Iron Overload and Iron Metabolism in Myelodysplastic Syndrome and Acute Myeloid Leukemia. Front Immunol. 2021 Feb 19;11:627662. doi: 10.3389/fimmu.2020.627662. PMID: 33679722; PMCID: PMC7933218.
- Cadet J, Wagner JR. DNA base damage by reactive oxygen species, oxidizing agents, and UV radiation. Cold Spring Harbor Perspect Biol (2013) 5(2):a012559. doi: 10.1101/cshperspect.a012559.
- Park S., Sapena R., Kelaidi C et al. Ferritin level at diagnosis is not correlated with poorer survival in non RBC transfusion dependent lower risk de novo MDS. Leuk. Res. 2011;35:1530–1533. doi: 10.1016/j.leukres.2011.07.007.
- Waszczuk-Gajda A., Madry K., Machowicz R. et al. Red blood cell transfusion dependency and hyperferritinemia are associated with impaired survival in patients diagnosed with myelodysplastic syndromes: Results from the first Polish MDS-PALG Registry. Clin. Exp. Med. 2016;25:633–641. doi: 10.17219/acem/62397.
Reviewer 2 Report
Dear authors, I enjoyed reading your paper as it is concise and well written.
However, in my opinion the manuscript is missing an overview of the role of IFI in the hematological patients unfit for chemotherapy but treated with other emerging therapies.
· I suggest to expand session (4. IRON OVERLOAD, IRON CHELATION THERAPY AND INVASIVE FUNGAL INFECTIONS IN HEMATOLOGY) adding data of unfit AML patients and patients with lymphoproliferative disorder
o (Aldoss I, Dadwal S, Zhang J, Tegtmeier B, Mei M, Arslan S, Al Malki MM, Salhotra A, Ali H, Aribi A, Sandhu K, Khaled S, Snyder D, Nakamura R, Stein AS, Forman SJ, Marcucci G, Pullarkat V. Invasive fungal infections in acute myeloid leukemia treated with venetoclax and hypomethylating agents. Blood Adv. 2019 Dec 10;3(23):4043-4049. doi: 10.1182/bloodadvances.2019000930. PMID: 31816059; PMCID: PMC6963254;
o Little JS, Weiss ZF, Hammond SP. Invasive Fungal Infections and Targeted Therapies in Hematological Malignancies. J Fungi (Basel). 2021 Dec 10;7(12):1058. doi: 10.3390/jof7121058. PMID: 34947040; PMCID: PMC8706272;
o Stefania Infante M, Fernández-Cruz A, Núñez L, Carpio C, Jiménez-Ubieto A, López-Jiménez J, Vásquez L, Del Campo R, Romero S, Alonso C, Morillo D, Prat M, Luis Plana J, Villafuerte P, Bastidas G, Bocanegra A, Serna Á, De Nicolás R, Marquet J, Mas-Ochoa C, Cordoba R, García-Suárez J, Comai A, Martín X, Bastos-Oreiro M, Seri C, Navarro-Matilla B, López-Guillermo A, Martínez-López J, Ángel Hernández-Rivas J, Ruiz-Camps I, Grande C; Grupo Español de Linfomas y Trasplante Autólogo de Medula Ósea (GELTAMO). Severe infections in patients with lymphoproliferative diseases treated with new targeted drugs: A multicentric real-world study. Cancer Med. 2021 Nov;10(21):7629-7640. doi: 10.1002/cam4.4293. Epub 2021 Sep 23. PMID: 34558211; PMCID: PMC8559487.)
· Please add consideration on primary prophylaxis of invasive fungal infections in patients with haematological malignancies
o Maertens JA, Girmenia C, Brüggemann RJ, Duarte RF, Kibbler CC, Ljungman P, Racil Z, Ribaud P, Slavin MA, Cornely OA, Peter Donnelly J, Cordonnier C; European Conference on Infections in Leukaemia (ECIL), a joint venture of the European Group for Blood and Marrow Transplantation (EBMT), the European Organization for Research and Treatment of Cancer (EORTC), the Immunocompromised Host Society (ICHS) and; European Conference on Infections in Leukaemia (ECIL), a joint venture of the European Group for Blood and Marrow Transplantation (EBMT), the European Organization for Research and Treatment of Cancer (EORTC), the Immunocompromised Host Society (ICHS) and the European LeukemiaNet (ELN). European guidelines for primary antifungal prophylaxis in adult haematology patients: summary of the updated recommendations from the European Conference on Infections in Leukaemia. J Antimicrob Chemother. 2018 Dec 1;73(12):3221-3230. doi: 10.1093/jac/dky286. PMID: 30085172.
o Mellinghoff SC, Panse J, Alakel N, Behre G, Buchheidt D, Christopeit M, Hasenkamp J, Kiehl M, Koldehoff M, Krause SW, Lehners N, von Lilienfeld-Toal M, Löhnert AY, Maschmeyer G, Teschner D, Ullmann AJ, Penack O, Ruhnke M, Mayer K, Ostermann H, Wolf HH, Cornely OA. Primary prophylaxis of invasive fungal infections in patients with haematological malignancies: 2017 update of the recommendations of the Infectious Diseases Working Party (AGIHO) of the German Society for Haematology and Medical Oncology (DGHO). Ann Hematol. 2018 Feb;97(2):197-207. doi: 10.1007/s00277-017-3196-2. Epub 2017 Dec 7. PMID: 29218389; PMCID: PMC5754425.
· Please consider to expand also the significance of iron overload and iron metabolism in myelodysplastic syndrome
o Weber S, Parmon A, Kurrle N, Schnütgen F, Serve H. The Clinical Significance of Iron Overload and Iron Metabolism in Myelodysplastic Syndrome and Acute Myeloid Leukemia. Front Immunol. 2021 Feb 19;11:627662. doi: 10.3389/fimmu.2020.627662. PMID: 33679722; PMCID: PMC7933218.
· Please add under both figures an additional short description of effects of iron overload on target organs and possible effects of iron chelation on fungal pathogens reported, clarifying the used abbreviations.
Author Response
Dear editor,
thank you, we accepted your suggestions.
We added under both figures a short description of effects of iron overload on target organs and possible effects of iron chelation on fungal pathogens reported, clarifying the used abbreviations:
Figure 1. Effects of iron overload on target organs. In affected organs, excess iron can chemically interact with hydrogen peroxide creating reactive oxygen species that can cause tissue damage, inflammation, and fibrosis. As such, iron overload can lead to cardiomyopathy, arrhythmias and heart failure, liver fibrosis and cirrhosis, diabetes mellitus, hypothyroidism, hypogonadism, and impotence. The impact of iron overload on the immune system and infections is less well understood, although there are indications that iron overload stimulates the growth and survival of some microorganisms and excessive amounts of iron in bone marrow stores act as an independent prognostic factor for invasive aspergillosis in allogenic transplant patients.
Figure 2. Possible effects of iron chelation on fungal pathogens. The use of iron chelators such as deferasirox increases the efficacy of antifungal drug Amphotericin B in invasive pulmonary aspergillosis and could reduce infectious complications in paediatric patients undergoing HSCT. Supplementation with calcium along with the use of iron chelators causes dramatic growth inhibition of the human fungal pathogens and increases efficacy of antifungal drugs.
We wrote an overview of the role of IFI in the hematological patients unfit for chemotherapy but treated with other emerging therapies, added information about primary prophylaxis of invasive fungal infections in patients with haematological malignancies, added data of unfit AML patients and patients with lymphoproliferative disorder and showed the significance of iron overload and iron metabolism in myelodysplastic syndrome.
However, we wanted to stay on topic with iron and invasive fungal infections, and not overexpand the section.
Please see added (bolded) text in the manuscript:
"There are several papers that investigated the role of IFI in the hematological patients unfit for chemoimmunotherapy but treated with other emerging therapies. Aldoss at al concluded that the overall risk of IFI during venetoclax and hypomethylating agents therapy is relatively low (12.6% patients developed probable or proven IFI in this investigation). The risk of IFI was higher in nonresponders, relapsed and refractory patients (1). There are real-world data suggesting a slightly higher risk of IFI in patients with chronic lymphocytic leukemia treated with ibrutinib (BTK inhibitor), especially in the first six months of treatment (2,3,4). The same could be true for PI3K inhibitors such as idelalisib (2,4,5). All this suggests the possible importance of antifungal prophylaxis in certain groups of patients. In principle, posaconazole remains the drug of choice when the incidence of invasive mold diseases exceeds 8%, and it’s strongly recommended for patients undergoing remission-induction chemotherapy for acute myeloid leukemia and myelodysplastic syndrome), as well as for preventing IFI in recipients of an allogenic hematopoietic stem cell transplantation, especially post-engraftment in the presence of graft-versus-host disease and other risk factors for IFI (6)."
"In their meta-analysis, Oliva and colleagues suggest that a higher concentration of serum ferritin is a prognostic indicator of shorter survival in patients with myelodysplastic syndrome, as shown by some earlier research (7, 8). Although some data suggest that non-transferrin-bound iron and labile plasma iron could have a proleukemic effect achieved through ROS (reactive oxygen species) (9, 10), the two papers included in this meta-analysis did not establish a possible relationship between serum ferritin level and progression to AML (11, 12)."
- Aldoss I, Dadwal S, Zhang J et al. Invasive fungal infections in acute myeloid leukemia treated with venetoclax and hypomethylating agents. Blood Adv. 2019 Dec 10;3(23):4043-4049. doi: 10.1182/bloodadvances.2019000930. PMID: 31816059; PMCID: PMC6963254.
- Stefania Infante M, Fernández-Cruz A, Núñez L et al., Grupo Español de Linfomas y Trasplante Autólogo de Medula Ósea (GELTAMO). Severe infections in patients with lymphoproliferative diseases treated with new targeted drugs: A multicentric real-world study. Cancer Med. 2021 Nov;10(21):7629-7640.
- Chamilos G, Lionakis MS, Kontoyiannis DP. Call for action: invasive fungal infections associated with ibrutinib and other small molecule kinase inhibitors targeting immune signaling pathways. Clin Infect Dis. 2018;66:140‐148.
- Little JS, Weiss ZF, Hammond SP. Invasive Fungal Infections and Targeted Therapies in Hematological Malignancies. J Fungi (Basel). 2021 Dec 10;7(12):1058. doi: 10.3390/jof7121058. PMID: 34947040; PMCID: PMC8706272).
- Brown JR, Byrd JC, Coutre SE et al. Idelalisib, an inhibitor of phosphatidylinositol 3-kinase p110δ, for relapsed/refractory chronic lymphocytic leukemia. Blood. 2014 May 29;123(22):3390-7. doi: 10.1182/blood-2013-11-535047. Epub 2014 Mar 10. PMID: 24615777; PMCID: PMC4123414.
- Maertens JA, Girmenia C, Brüggemann RJ et al. European Conference on Infections in Leukaemia (ECIL), a joint venture of the European Group for Blood and Marrow Transplantation (EBMT), the European Organization for Research and Treatment of Cancer (EORTC), the Immunocompromised Host Society (ICHS) and the European LeukemiaNet (ELN). European guidelines for primary antifungal prophylaxis in adult haematology patients: summary of the updated recommendations from the European Conference on Infections in Leukaemia. J Antimicrob Chemother. 2018 Dec 1;73(12):3221-3230. doi: 10.1093/jac/dky286. PMID: 30085172.
- Oliva EN, Huey K, Deshpande S et al. A Systematic Literature Review of the Relationship between Serum Ferritin and Outcomes in Myelodysplastic Syndromes. J Clin Med. 2022 Feb 8;11(3):895. doi: 10.3390/jcm11030895. PMID: 35160344; PMCID: PMC8836890.
- Malcovati L, Della Porta MG, Cazzola M. Predicting survival and leukemic evolution in patients with myelodysplastic syndrome. Haematologica. 2006 Dec;91(12):1588-90. PMID: 17145593.
- Weber S, Parmon A, Kurrle N, Schnütgen F, Serve H. The Clinical Significance of Iron Overload and Iron Metabolism in Myelodysplastic Syndrome and Acute Myeloid Leukemia. Front Immunol. 2021 Feb 19;11:627662. doi: 10.3389/fimmu.2020.627662. PMID: 33679722; PMCID: PMC7933218.
- Cadet J, Wagner JR. DNA base damage by reactive oxygen species, oxidizing agents, and UV radiation. Cold Spring Harbor Perspect Biol (2013) 5(2):a012559. doi: 10.1101/cshperspect.a012559.
- Park S., Sapena R., Kelaidi C et al. Ferritin level at diagnosis is not correlated with poorer survival in non RBC transfusion dependent lower risk de novo MDS. Leuk. Res. 2011;35:1530–1533. doi: 10.1016/j.leukres.2011.07.007.
- Waszczuk-Gajda A., Madry K., Machowicz R. et al. Red blood cell transfusion dependency and hyperferritinemia are associated with impaired survival in patients diagnosed with myelodysplastic syndromes: Results from the first Polish MDS-PALG Registry. Clin. Exp. Med. 2016;25:633–641. doi: 10.17219/acem/62397.